# HIV Drug Resistance Patterns and Characteristics Associated with Clinically Significant Drug Resistance among Children with Virologic Failure on Antiretroviral Treatment in Kenya: Findings from the Opt4Kids Randomized Controlled Trial

**DOI:** 10.3390/v15102083

**Published:** 2023-10-12

**Authors:** Lisa Abuogi, Patrick Oyaro, Garoma Wakjira, Katherine K. Thomas, Andrea J. Scallon, Irene Mukui, Bhavna H. Chohan, Evelyn Brown, Enericah Karauki, Nashon Yongo, Bilaal Ahmed, Shukri A. Hassan, James Wagude, Eunice Kinywa, Linda Otieno, Leonard Kingwara, Boaz Oyaro, Lisa M. Frenkel, Grace John-Stewart, Rena C. Patel

**Affiliations:** 1Department of Pediatrics, University of Colorado, Denver, CO 80045, USA; 2Health Innovations Kenya (HIK), Kisumu 40100, Kenya; patrickoyaro@gmail.com; 3United States Agency for International Development, Washington, DC 20004, USA; 4Department of Medicine, University of Washington, Seattle, WA 98195, USA; garo20uw@uw.edu (G.W.); shukrih@uw.edu (S.A.H.); lfrenkel@uw.edu (L.M.F.); gjohn@uw.edu (G.J.-S.); renapatel@uabmc.edu (R.C.P.); 5Department of Global Health, University of Washington, Seattle, WA 98105, USA; kkthomas@uw.edu (K.K.T.); ascallon@uw.edu (A.J.S.); bchohan@uw.edu (B.H.C.); 6Drugs for Neglected Diseases Initiative (DNDI), Nairobi 21936, Kenya; irenemukui5@gmail.com; 7Kenya Medical Research Institute, Nairobi 00200, Kenya; 8UWKenya, Nairobi 00200, Kenya; 9Department of Health, Ministry of Health, Siaya 40600, Kenya; 10Department of Health, Ministry of Health, Kisumu 40100, Kenya; 11Family AIDS Care and Education Services, Kenya Medical Research Institute, Kisumu 40100, Kenya; 12National HIV Reference Laboratory, Kenya Ministry of Health, Nairobi 00202, Kenya; leonard.kingwara@gmail.com; 13Kenya Medical Research Institute-CDC, Kisumu 40100, Kenya; boyaro@kemri.go.ke; 14Departments of Pediatrics, University of Washington, Seattle, WA 98195, USA; 15Department of Laboratory Medicine and Pathology, University of Washington, Seattle, WA 98195, USA

**Keywords:** human immunodeficiency virus (HIV), children, drug resistance, virologic failure

## Abstract

Increasing HIV drug resistance (DR) among children with HIV (CHIV) on antiretroviral treatment (ART) is concerning. CHIV ages 1–14 years enrolled from March 2019 to December 2020 from five facilities in Kisumu County, Kenya, were included. Children were randomized 1:1 to control (standard-of-care) or intervention (point-of-care viral load (POC VL) testing every three months with targeted genotypic drug resistance testing (DRT) for virologic failure (VF) (≥1000 copies/mL)). A multidisciplinary committee reviewed CHIV with DRT results and offered treatment recommendations. We describe DR mutations and present logistic regression models to identify factors associated with clinically significant DR. We enrolled 704 children in the study; the median age was 9 years (interquartile range (IQR) 7, 12), 344 (49%) were female, and the median time on ART was 5 years (IQR 3, 8). During the study period, 106 (15%) children had DRT results (84 intervention and 22 control). DRT detected mutations associated with DR in all participants tested, with 93 (88%) having major mutations, including 51 (54%) with dual-class resistance. A history of VF in the prior 2 years (adjusted odds ratio (aOR) 11.1; 95% confidence interval (CI) 6.3, 20.0) and less than 2 years on ART at enrollment (aOR 2.2; 95% CI 1.1, 4.4) were associated with increased odds of major DR. DR is highly prevalent among CHIV on ART with VF in Kenya. Factors associated with drug resistance may be used to determine which children should be prioritized for DRT.

## 1. Introduction

Children living with human immunodeficiency virus (HIV) (CHIV) currently require life-long antiretroviral therapy (ART) and continue to have more limited medication options than adolescents and adults living with HIV [1,2]. Additionally, children are frequently exposed to antiretrovirals (ARVs) as part of perinatal transmission prevention efforts. Rates of virologic suppression in CHIV on ART are lower than for adults on treatment, exposing children to consequent risks of poor growth, suboptimal neurodevelopment, clinical progression including opportunistic infections, and mortality [3].

The World Health Organization (WHO) and others have expressed concerns that HIV drug resistance (DR) to ARVs will undermine the attainment of the global targets for HIV, and CHIV face unique challenges for DR [4,5,6,7,8]. Pre-treatment DR in ART-naïve infants living with HIV is alarmingly high with a pooled estimate of 45.5% from recent multi-country reports and is associated with increased risk of virologic failure (VF) [6]. Acquired DR while on ART is also increasingly recognized as a concern among CHIV, but less data are available. Among children with virologic failure (VF), small surveillance reports in east and southern Africa indicate that between 48–71% may have DR mutations to non-nucleos(t)ide-reverse transcriptase inhibitor (NNRTI)-containing ART [6,9,10]. This problem is likely to be worsening over time, with more ART exposure through earlier initiation and combination ART now being used for the prevention of perinatal transmission [11,12]. Despite the recent rollout of integrase inhibitor regimens, primarily dolutegravir (DTG)-containing regimens, for CHIV, DR is likely to be a continued threat to the sustainable use of these regimens and the impact of common NRTI DR in CHIV is not fully understood [13,14,15].

Kenya is heavily burdened by HIV, with an estimated 68,000 CHIV with around 85% on ART in 2022 [16]. A national pediatric HIV DR surveillance study conducted in 2013 revealed that at least a third of children had VF on first-line ART and over 90% of those had multiple DR [17]. This survey, however, was conducted prior to the use of more efficacious ART regimens among CHIV, including protease inhibitor (PI)-containing ART, and universal ART for all pregnant women [17]. The current levels and pattern of DR among CHIV with VF in Kenya are not known.

Understanding DR patterns and associated factors among children on ART in Kenya may inform algorithms used to manage children with VF. Here, we present DR testing (DRT) results from a randomized trial evaluating targeted DRT for children with VF accompanied by a collaborative, multidisciplinary case review to inform ART recommendations. We describe the drug resistance patterns, characteristics associated with major drug resistance, and clinical outcomes of CHIV undergoing DRT from this trial.

## 2. Materials and Methods

The Optimizing Viral Suppression in Children on ART in Kenya (Opt4Kids) study protocol and primary findings have been described previously [18,19]. In brief, 704 CHIV ages 1–14 years were enrolled from 5 public facilities in Kisumu County, Kenya, between March and December 2019 and were followed for 12 months. Children were individually randomized 1:1, stratified by site and age groups (ages 1–9 years and 10–14 years), to the control (standard-of-care) or intervention groups (point-of-care viral load testing every three months with DRT for those with VF (HIV RNA ≥ 1000 copies/mL)) and followed for 12 months. Participants in both groups underwent point-of-care (POC) viral load (VL) testing and targeted DRT, if indicated, at 12 months post-enrollment.

### 2.1. Study Procedures

Whole blood was collected from study participants for POC VL testing and separated into plasma for testing using a GeneXpert system (Cepheid, Nairobi, Kenya) on site at study facilities or via daily transport to a facility less than 2 km away [20]. The study facilities participated in a quarterly external quality assurance program for HIV POC VL testing using GeneXpert, and each facility passed each check. HIV DR testing was performed on plasma samples using Sanger sequencing with Applied Biosystems 3130xl Genetic Analyzers (ThermoFisher Scientific, Nairobi, Kenya) at the KEMRI-CDC HIV Research and Sanger 3730xl at the Kenya National HIV Reference Laboratories. These laboratories utilize validated, WHO-certified, optimized in-house assays to detect reverse transcriptase and PI mutations [21,22]. Integrase strand transfer inhibitor (INSTI) mutations were not routinely evaluated during the study period. The DRT result reports contained a list of the DR genotypes as well as phenotypic interpretations based on the scoring systems generated by the Stanford Genotypic Resistance Interpretation Algorithm (Stanford Univeristy, Palo Alto, CA, USA) versions available during the study period [23].

Children in the intervention group underwent DRT at episodes of VF detection. A multidisciplinary committee, called the Clinical Management Committee (CMC), which included facility providers and peer leaders, HIV implementing partner technical advisors, the chairperson of the local HIV Technical Working Group, and study principal investigators, reviewed the DRT results to provide interpretation and clinical management recommendations. The CMC utilized a standardized Kenya Ministry of Health (MoH) case summary form, prepared by facility providers and research staff in advance of the CMC review, to discuss the cases. The CMC met as the DRT results were available, often weekly, throughout the study period. The CMC recommendations were summarized orally during virtual meetings and shared in writing along with the DRT results to facility providers within one week.

For children in the control group, providers were instructed to follow current Kenya MoH guidelines for the management of any child with VF, which recommended enhanced adherence counseling for children with VF and repeat viral load (VL) testing after three months of provider-determined “good” adherence. DRT was limited to patients approved by the regional HIV Technical Working Group and generally included those failing a PI-containing regimen or with persistent VF despite good adherence. The Technical Working Group reviewed case summaries and DRT results, when available, and provided guidance to facility staff on patient management, though facility staff did not participate in working group meetings. In practice, only two control group participants had DRT results which were received after the study end. Therefore, DRT results from control group participants prior to the final study visit were excluded from this analysis. At the study end, participants in the control group underwent intervention procedures, including CMC review for those with DRT results.

### 2.2. Study Setting

First-line ART regimens in Kenya during the study period for children included lamivudine with either abacavir (preferred) or zidovudine (alternative) and lopinavir/ritonavir for those less than three years of age and lamivudine with either abacavir (preferred) or zidovudine (alternative) and efavirenz for those three years of age and older [24]. Second-line ART regimens included changes in NRTI medication from abacavir to zidovudine or vice versa depending on which the child was on as first-line with maintenance of lamivudine. Those on lopinavir/ritonavir as first-line required review by the HIV Technical Working Group to recommend a second-line regimen, while those on efavirenz were recommended to switch to lopinavir/ritonavir. In 2020, the guidelines were updated to recommend dolutegravir (DTG) for those weighing at least 20 kg for treatment initiation and optimization (switch to DTG-containing regimen regardless of viral load), which was rolled out during the study period [25].

### 2.3. Study Population

Children 1–14 years of age, enrolled at a study site, on or initiating ART, and with a consenting caregiver were enrolled to the study. Participant characteristics at enrollment have been previously published [26]. Of note, 536 (76%) were virally suppressed, 77 (11%) had VF, 91 (13%) had missing VL data, and 20 participants newly initiated ART within 30 days of study enrollment. HIV care and treatment were provided by government staff as per national guidelines.

### 2.4. Data Collection and Management

We abstracted routinely collected data from standardized Ministry of Health forms in medical files and registers using direct, electronic data entry via tablets into a REDCap database. Similarly, we entered study-collected data, including DRT results, in this REDCap database.

### 2.5. Primary Analytic Outcome

A participant was considered to have clinically significant DR if they had any mutation listed for NRTI and NNRTI drugs with a penalty score or if listed as “major” for PIs by Stanford’s Genotypic Resistance Interpretation Algorithm (i.e., Stanford HIVdb) on any DRT [23].

### 2.6. Exposures and Covariates

We selected potential risk factors a priori, based on the existing literature and content knowledge, which included age, sex, duration on ART, prior ARV exposure, and prior history of VF [27,28]. History of VF was defined as any VL result ≥ 1000 copies/mL within two years prior to study enrollment. Given the exploratory nature of this analysis, we also included the following covariates: WHO stage, self-reported HIV status of primary caregiver at enrollment, socioeconomic status (household commodities ownership), urban, peri-urban, or rural clinic location, and clinic volume defined as high, medium, and low based on the number of HIV patient visits per month at each facility.

### 2.7. Statistical Analysis

First, we describe the proportion of participants from either group who underwent DRT as part of the study intervention or at the study end, per protocol, and the proportion with DR mutations detected by HIV drug classes, e.g., NRTIs, NNRTIs, and PIs. All DRTs for each participant were reviewed, but only one DRT result contributed to the data analysis. For a report of DRMs by drug class, the DRT with the highest number of mutations was used. We report on the prevalence of HIV DR by ARV medication using WHO’s definition of a penalty score of ≥15 using the Standford HIVdb algorithm [6,23]. We estimated the prevalence of the most common DR mutations (K65R, L74V/I, Y115F, M184V, K103N, Y181C, G190A) with 95% confidence intervals (CI).

Second, to evaluate which CHIV were most likely to have clinically relevant DR, we used logistic regression models to identify factors associated with clinically significant DR. Variables with *p*-values < 0.20 in univariate analyses were included in the multivariate model.

Third, to explore any potential impact of DRT on clinical outcomes, we report descriptive statistics for outcomes including viral suppression (VS), loss to follow up, and death by study group among those with DRT results, categorized by whether an ART regimen change occurred or not.

## 3. Results

A total of 704 children were enrolled in the study with a median age 9 years (interquartile range (IQR) 7, 12); 344 (49%) were female, and the median time on ART was 5 years (IQR 3, 8). A total of 349 (49.5%) and 355 (50.5%) of the CHIV were randomized to the intervention and control groups, respectively. Overall, 382 (54.3%) of the participants were on an NNRTI-containing regimen at study enrollment, 294 (41.8%) were on a PI-containing regimen, 27 (3.8%) were on an INSTI-containing regimen, and 1 (0.1%) was on a PI and INSTI regimen (Table 1).

During the 12-month study period, 190 study DRTs were requested for 106 (15%) participants across the two study groups. Among the intervention participants, 88 (25%) children experienced at least one episode of VF, and all had a least one DRT requested for a total of 166 DRT requests, and 152 (92%) had results. (Figure 1) A total of 66 (19%) children in the control group had a least one episode of viremia. as per the study protocol, those with VF identified at the 12-month study visit were included in the analysis (*n* = 24), all of whom had a study DRT requested, of which 22 (92%) had results.

Of the total 190 DRTs requested by the study, 16 (9%) total samples from 16 participants did not yield a result, with 14 (87%) samples failing to amplify and 2 (13%) having insufficient volume to be tested. Fourteen participants in the intervention group had a “failed” result, of whom ten (71%) had a successful subsequent repeat DRT. An additional three (21%) intervention participants re-suppressed to VL < 1000/mL and had no further indication for DR testing. Two children in the control group who had DRT performed at 12-month visit per protocol that failed to amplify were referred for repeat of DRT by facility staff via routine care.

### 3.1. Drug Resistance among Children on ART with Virologic Failure

Among the one hundred and six participants with at least one DRT result, all demonstrated at least one clinically significant mutation or minor DR mutation, as defined by the Stanford HIV Database. A total of 93 (87.7%) had clinically significant mutations, and 13 (12.3%) had minor mutations only (Table 1). In the 93 children with clinically significant resistance, 87 (93.5%) had NNRTI resistance, 70 (75.3%) had NRTI resistance, and 10 (10.8%) had PI resistance. Additionally, more than half of the children with significant resistance had dual-class resistance to NRTI and NNRTIs (*n* = 51, 54.8%), two (2.2%) had dual-class resistance with either NRTI or NNRTI and PIs, and eight (8.6%) had triple-class resistance to NRTI, NNRTI, and PIs. The most common DR mutations identified were among NRTI and NNRTIs, with most common being M184V (57.5%, 95% CI 48%, 67%) followed by K103N (35.8%, 95% CI 27%, 45%) (Table 2). Additional descriptions of DR by drug class are shown in Figure 2. Over half of those with DRT results demonstrated DR to abacavir, emtricitabine, lamivudine, efavirenz, and nevirapine, while over 10% had resistance to zidovudine and tenofovir (Figure 3). DR to newer-generation NNRTIs was also high, with 43.5% having DR to rilpivirine and fewer having DR to etravirine (66.0%) and doravirine (20.0%).

Among the eighty-eight CHIV in the intervention group of the study who had DRT requested, forty-seven (53%) had more than one DRT requested for repeat viremia during the study, whereas no CHIV in the control group had more than one DRT. Only three participants initially had minor resistance and later had clinically significant resistance on repeat DRT. Similarly, only six children went from a single class with resistance to two or more drug classes. No children with a change in resistance on follow-up DRT had ART regimen changes between DRTs.

### 3.2. Characteristics Associated with Major Drug Resistance

The associations between participant characteristics and clinically significant DRs are shown in Table 3. Younger children 1–4 years of age had three-fold higher odds of clinically significant DR compared to older children aged 5–10 years or 11–14 years demonstrated (odds ratio (OR) 3.2, 95% CI 1.6, 6.1). Similarly, those on ART for <2 years and children on a PI-containing regimen at enrollment or first DRT had higher odds of major DR (OR 2.9, 95% CI 1.7–4.9, OR 1.9, 95% CI 1.2, 2.0, and OR 1.9, 95% CI 1.2,3.0, respectively) (Table 3). Further, a history of virologic failure in the 2 years prior to study enrollment was associated with a nearly 11-fold increased odds of major DR compared to no prior VF (95% CI 6.6, 18.5). In an adjusted analysis, time on ART and history of virologic failure remained significantly associated with major DR. There were no associations with the type of primary caregiver (e.g., mother, father, other family, or non-family member), the HIV status of the primary caregiver, household socioeconomic status, or facility volume or location between groups.

### 3.3. Clinical Management and Outcomes of Children with DRT

The CMC carried out case reviews for all participants with DRT results and recommended an ART regimen change for 46 (43%) out of the 106 participants with a DRT. In the control group, 22 participants had a DRT after the 12-month study visit, and 100% had any DR with 19 (86%) with major DR. Eight of those with results (36.4%) had a recommendation for an ART regimen change after clinical review. The control group participants were not followed beyond this final study visit.

In the intervention group, 38/88 (43.2%) were recommended to change ART after a case review, while 34 (38.6%) were advised to continue the current regimen. An additional 12 (13.6%) intervention participants underwent an ART change due to a national transition to dolutegravir that was initiated by the clinic providers. Of those with an ART change recommendation in the intervention group, 35 (92.1%) had changed to the recommended ART by the study end. All the intervention participants with an ART change recommendation had clinically significant DR. Nearly a third (*n* = 11, 28.9%) with an ART change recommendation were advised to change the non-NRTI ARV but were able to preserve their current NRTI medications based on DRT results, which would have otherwise been switched as per in-country guidelines. A total of 31/34 (91.2%) children who did not have an ART change recommendation had major DR, but their current regimen was still evaluated to be effective. Most of the children 27/34 (79.4%) without an ART regimen change recommendation were on a PI-containing regimen without major PI resistance and with no age-appropriate available medication alternatives (e.g., dolutegravir formulation not available or recommended for age at time of study). An additional three older participants were on an INSTI-containing regimen (specifically dolutegravir), and four were on an NNRTI (specifically efavirenz) without DR impacting the current regimen.

For the intervention participants with DRT prior to the final study visit (*n* = 77), 32 (42%) were virally suppressed at the study end (Table 4). Viral suppression (VS) at the 12-month visit was observed in 21/36 (58%) of patients with a recommendation to change ART and in 12/31 (39%) recommended to not change ART (*p* = 0.40). Additionally, VS was observed in 9/10 (90%) of patients with a programmatic switch to DTG. Loss to follow up was higher in those without an ART change recommendation (*n* = 6, 19%) compared to those with an ART change recommendation (*n* = 3, 8%) (*p* = 0.28). One child died in the study who had VF and a DRT result that did not result in an ART change recommendation.

In this cohort of over 700 children on ART in Kenya, we detected high levels of DR in CHIV with VF. All children with VF had some DR, the majority of whom had major DR. Children on ART for >2 years and those with a history of VF were significantly more likely to have major DR. We observed that half of the CHIV with DR re-suppressed by the study end following the use of a multidisciplinary care team approach for case review and clinical management recommendations. Our study provides data on evolving DR patterns in CHIV and can inform prioritization approaches of DRT for vulnerable groups of CHIV as new ARV options become available.

## 4. Discussion

Our study identified major DR in most CHIV with VF. The last published comprehensive DR surveillance for children in Kenya was in 2013 before the changes to the recommend PI-containing ART for children less than three years of age came into effect [17]. Over 90% of the children in this national assessment were prescribed an NNRTI-containing regimen compared to less than 50% in our contemporary cohort. However, a similarly high proportion of CHIV with treatment failure were found to have any DR (89%) in the national survey. More recent surveillance studies carried out in children in Lesotho, Uganda, and Zambia demonstrate high rates of NRTI (50–80%) and NNRTI (84–97%) resistance and low rates of PI resistance (4–6%) in children with VF. Related adult HIV DR surveillance studies carried out in eight African countries between 2015 and 2019 also show high rates of NRTI and NNRTI DR among those with VF on an NNRTI-containing regimen. The low rates of PI resistance in children with VF show that despite the transition to PI-containing regimens for a significant proportion of children, PI medications (specifically lopinavir/ritonavir among our cohort) maintained a high barrier to resistance. However, PI-containing regimens remain less attractive than alternatives; adult surveillance studies and one study in children from Zambia consistently identify lower viral suppression among people on PI-containing regimens compared to INSTI- or NNTRI-containing regimens, likely due to non-adherence [6]. Additionally, the need to dose twice daily and the side effect profile of lopinavir/ritonavir support the transition away from this PI drug in children when other options are available.

Developing strategies to optimize cost-effective use for targeted DRT among CHIV with VF is key to achieving higher rates of viral suppression in CHIV. A prior history of VF was a strong predictor of major DR. This finding is important, as the guidelines in many countries do not take this into account in their algorithms to guide the management of children with VF. Studies in both adults and children have demonstrated the accumulation of new drug resistance with continued virologic failure [8,29]. Additionally, children on ART for less than 2 years with VF had higher odds of major DR compared to CHIV on ART for longer periods. It is possible that children on ART for shorter periods were younger children who received antiretroviral prophylaxis during breastfeeding selecting for pretreatment DR [7]. Pre-treatment DR is a growing concern noted by the WHO and others, and NNRTI resistance has increased to a pooled estimate of 46% in infants, whereas it is less than 15% in newly initiating adults [6]. We had very few children newly initiating ART and so were unable to estimate pre-treatment DR in this cohort.

Our findings support the broader use of DRT in CHIV, which is lacking in many settings [30]. While many HIV providers in low- and middle-income countries (LMICs) lack experience interpreting DRT, our multidisciplinary virtual approach to case reviews was feasible and highly acceptable. Given that nearly 60% of the children with prior VF in the last two years demonstrated major DR in our study, guidelines should consider if DRT should be utilized sooner in children with persistent VF since adherence interventions alone may be insufficient. DRT may also allow for the preservation of NRTIs in children with VF, which is important given the limited ARV options for CHIV and the potential side effects of less preferred NRTIs such as zidovudine. Even when major DR is not identified by DRT, these findings can inform clinic staff of the need to re-focus on the psychosocial and structural factors contributing to non-adherence.

The rapid transition to DTG-containing regimens in children provides hope for improved VS in all CHIV, including for those with VF. Our findings showed high viral resuppression in a small number of children with VF who were switched to a DTG-containing regimen. How DRT should be used in the setting of wide-spread DTG use in children remains uncertain. NRTI mutations were highly prevalent in the CHIV with VF in this cohort, which is similar to other reports in children [6]. Over half of the children in our cohort were resistant to abacavir and lamivudine and over 10% were resistant to tenofovir. This is concerning given that these drugs are part of the current first-line regimens co-formulated with DTG for CHIV in Kenya and other LMICs. While some adult studies show that DTG-containing regimens may remain suppressive even in the setting of NRTI DR, this remains to be demonstrated in CHIV [31,32]. Kenya’s current treatment guidelines recommend switching to PI-containing regimens for children who fail on DTG, but as noted in our study and others, those on PI-containing regimens as a second line were less likely to resuppress [6]. Kenya has incorporated recommendations for DRT in CHIV who experience VF on DTG, but subsequent regimen sequencing is uncertain, especially in light of frequent NRTI DR. Our results also show a concerningly high level of DR to rilpvirine, threatening the use of the highly anticipated long-acting injectable cabotegravir/rilpivirine for CHIV. Similar results in adults in South Africa and other settings have prompted investigators to suggest that DRT may be required before long-acting injectable ART can be used in LMICs [33,34]. Thus, there is likely an important role for targeted DRT in determining how best to manage CHIV with VF while on DTG.

The limitations of this study include the inability to determine pre-treatment DR due to the fact that there were few children who were initiating ART in this cohort and the lack of data on maternal ART and DR history, which is likely to correlate with DR identified in participants. Additionally, our study utilized DRT laboratories in Kenya to optimize the generalizability of DRT use in CHIV, but INSTI DR testing was limited during the study period. While we observed a rapid transition to DTG-containing regimens, we were not able to perform INSTI DR on CHIV with VF on DTG. We plan to use stored samples to further explore these issues.

## 5. Conclusions

The findings from this study demonstrated high levels of major DR in children living with VF. Providers and policy makers should consider the identified factors associated with major DR when considering which children may benefit most from DRT while it remains a limited resource. Further research is needed to understand how DRT may be optimally used among children who are now mostly using DTG-containing regimens.

## Figures and Tables

**Figure 1 viruses-15-02083-f001:**
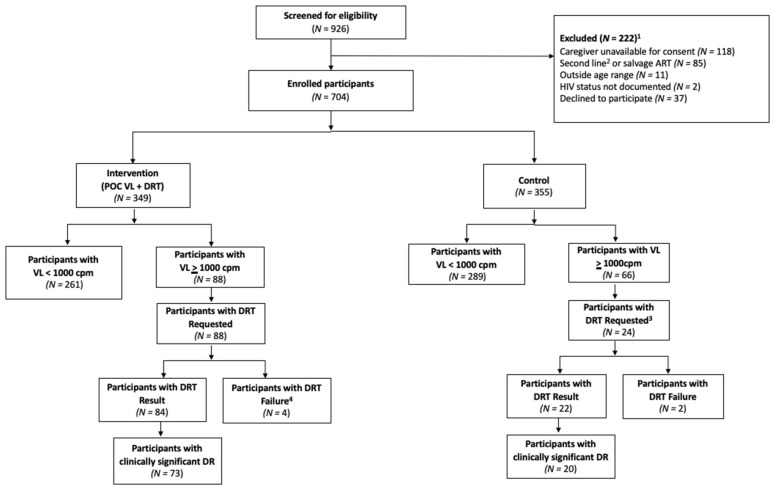
Opt4Kids study participants undergoing drug resistance testing by study group. ^1^ Note, categories for exclusion not mutually exclusive. ^2^ Protocol changed to allow for enrollment of second-line ART. ^3^ DRT requests per study protocol only. ^4^ There were additional DRT failures, but those participants had at least one successful DRT. See text for details. ART—antiretroviral treatment; POC—point of care; VL—viral load; DR—drug resistance; DRT—drug resistance test; cpm—copies per milliliter.

**Figure 2 viruses-15-02083-f002:**
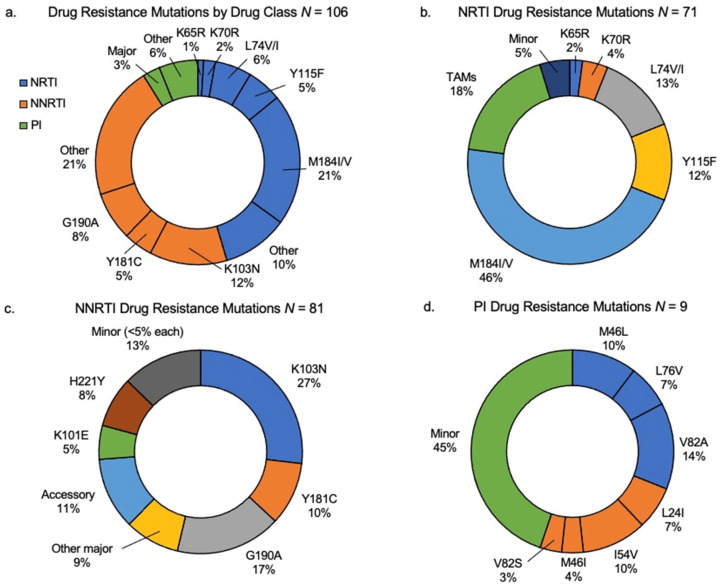
Description of HIV drug resistance mutations among children on ART with virologic failure. *N* = number of children with resistance in the respective drug class(es). (**a**) Describes overall drug resistance by drug class and select codons in all children undergoing drug resistance testing; (**b**) prevalence of NRTI mutations conferring high (major)- or low (minor)-level drug resistance; (**c**) prevalence of select mutations conferring NNRTI drug resistance; (**d**) prevalence of mutations characterized as major (blue), other major (orange), or minor (green) drug resistance to PI drugs. NRTIs—nucleoside reverse transcriptase inhibitors; NNRTIs—non-nucleoside reverse transcriptase inhibitors; PIs—protease inhibitors.

**Figure 3 viruses-15-02083-f003:**
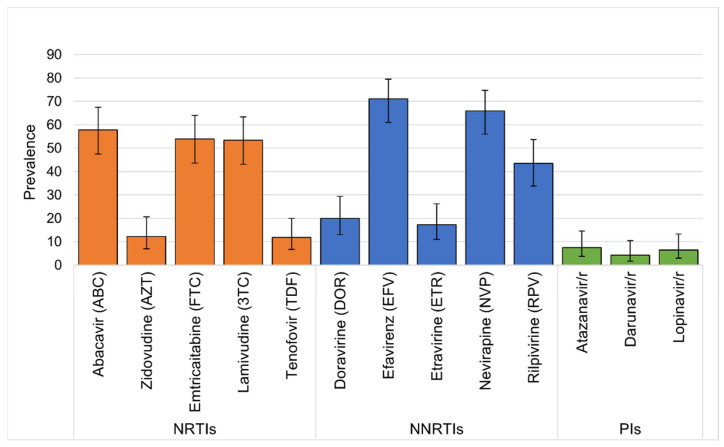
Prevalence (and 95% confidence intervals) of HIV drug resistance by antiretroviral medication among children with virologic failure. NRTIs—nucleoside reverse transcriptase inhibitors; NNRTIs—non-nucleoside reverse transcriptase inhibitors; PIs—protease inhibitors.

**Table 1 viruses-15-02083-t001:** Characteristics of children enrolled in Opt4Kids by detection of HIV drug resistance mutations.

	Total n = 704	Children without Any Drug Resistance Testing n = 598	Children with DRT	
			Children with Any Drug Resistance n = 106	Children with Clinically Significant Drug Resistance Only n = 93
Characteristics	N (%) ^1^	N (%)	N (%)	N (%)
Age, median (IQR)	9 (7, 12)	9 (7, 12)	9 (4, 12)	9 (4, 12)
Age category (years)				
1–4	55 (7.8)	34 (5.7)	21 (19.8)	17 (18.3)
5–10	325 (46.2)	284 (47.5)	41 (38.7)	36 (38.7)
11–14	324 (46.0)	280 (46.8)	44 (41.5)	40 (43.0)
Sex				
Male	360 (51.1)	311 (52.0)	49 (46.2)	42 (45.2)
Female	344 (48.9)	287 (48.0)	57 (53.8)	51 (54.8)
Time on ART (years) median (IQR)	5 (3, 8)	6 (3, 8)	4 (1, 8)	4 (1, 8)
Time on ART categorical				
<2 years	110 (15.6)	76 (12.7)	34 (32.1)	29 (31.2)
2–5 years	186 (26.4)	162 (27.1)	24 (22.6)	18 (19.4)
>5 years	400 (56.8)	354 (59.2)	46 (43.4)	44 (47.3)
Missing	8 (1.1)	6 (1.0)	2 (1.9.0)	2 (2.2)
ART regimen at enrollment				
NNRTI-based	382 (54.3)	336 (56.2)	46 (43.4)	38 (40.9)
PI-based	294 (41.8)	238 (39.8)	56 (52.8)	51 (54.8)
Integrase-based	27 (3.8)	23 (3.8)	4 (3.8)	4 (4.3)
Other	1 (0.1)	1 (0.2)	0	0
NRTI ART regimen at enrollment				
ABC+3TC	498 (70.7)	422 (70.6)	76 (71.7)	64 (68.8)
AZT+3TC	113 (16.1)	93 (15.6)	20 (18.9)	19 (20.4)
TDF+3TC	93 (13.2)	83 (13.8)	10 (9.4)	10 (10.8)
ART regimen at 1st DRT				
NNRTI-based	N/A	N/A	46 (43.4)	38 (40.9)
PI-based			56 (52.8)	51 (54.8)
INSTI-based			4 (3.8)	4 (4.3)
WHO stage				
1 or 2	487 (69.2)	410 (68.6)	77 (72.6)	67 (72.0)
3 or 4	152 (21.6)	134 (22.4)	18 (17.0)	17 (18.3)
Missing	65 (9.2)	54 (9.0)	11 (10.4)	9 (9.7)
History of virologic failure within 2 years prior to study				
Yes	144 (20.5)	87 (14.5)	57 (53.8)	55 (59.1)
No	504 (71.6)	470 (78.6)	34 (32.0)	27 (29.0)
Missing	56 (8.0)	41 (6.9)	15 (14.2)	11 (11.8)
Primary Caregiver				
Mother	482 (68.4)	409 (68.4)	73 (68.9)	63 (67.7)
Father	59 (8.4)	53 (8.9)	6 (5.7)	5 (5.4)
Other biological	124 (17.6)	107 (17.9)	17 (16.0)	17 (18.3)
Other non-biological	39 (5.5)	29 (4.8)	10 (9.4)	8 (8.6)
Primary caregiver HIV status				
Positive	568 (80.7)	485 (81.1	83 (78.3)	74 (79.8)
Negative	135 (19.2)	112 (18.7)	23 (21.7)	19 (20.2)
Unknown	1 (0.1)	1 (0.2)	0	0
Household commodities ownership				
Electricity	403 (57.2)	346 (57.9)	57 (53.8)	50 (53.8)
Radio	554 (78.7)	472 (78.9)	82 (77.4)	73 (78.5)
Television	375 (53.3)	318 (53.2)	57 (53.8)	52 (56.0)
Phone	679 (96.4)	577 (96.4)	101 (95.3)	89 (95.7)
More than one room	580 (82.4)	494 (82.6)	86 (81.1)	74 (79.6)
Clinic volume				
Heavy	421 (59.8)	362 (60.5)	59 (55.7)	55 (59.0)
Medium	158 (22.4)	130 (21.7)	28 (26.4)	24 (26.0)
Light	125 (17.8)	106 (17.7)	19 (18.0)	14 (15.0)
Clinic location				
Urban	421 (59.8)	362 (60.5)	59 (55.7)	55 (59.1)
Seri-urban	158 (22.4)	130 (21.7)	28 (26.4)	24 (25.8)
Rural	125 (17.8)	106 (17.7)	19 (18.0)	14 (15.1)

**^1^** Number (N) and percent (%) except where indicated.

**Table 2 viruses-15-02083-t002:** Prevalence and 95% confidence intervals for most common HIV drug resistance mutations detected among children with virologic failure undergoing drug resistance testing in the Opt4Kids trial, March 2019–April 2021.

Drug Class	Drug Resistance Mutation	CHIV with DRM *n* = 106 *N*	Prevalence (%) (95% Confidence Interval)
Nucleoside reverse transcriptase inhibitor (NRTIs)			
	K65R	3	2.8 (0.01, 0.08)
	Y115F	17	16.0 (0.10, 0.24)
	L74V/I	18	17.0 (0.11, 0.25)
	M184V	61	57.5 (0.48, 0.67)
	Total NRTI	99	93.4 (0.87, 0.97)
Non-nucleoside reverse transcriptase (NNRTIs)			
	Y181C	14	13.2 (0.08, 0.21)
	G190A	25	23.6 (0.17, 0.33)
	K103N	38	35.8 (0.27, 0.45)
	Total NNRTI	77	72.6 (0.63, 0.80)

**Table 3 viruses-15-02083-t003:** Factors associated with major HIV drug resistance mutations among children with drug resistance testing.

Characteristics	Children with Major Drug Resistance (*n* = 93)	Children without Major Drug Resistance (*n* = 611)	Unadjusted OR (95% CI)	*p*-Value	Adjusted OR (95% CI)	*p*-Value
Age of child ^1^						
1–4	17 (31.0)	38 (69.0)	3.18 (1.61, 6.10)	<0.001	2.01 (0.84, 4.76)	0.113
5–10	36 (11.1)	289 (89.0)	0.88 (0.55, 1.43)	0.615	0.84 (0.48, 1.46)	0.534
11–14 (ref)	40 (12.3)	284 (87.7)	1		1	
Gender						
Male (ref)	42 (11.7)	318 (88.3)	1			
Female	51 (14.8)	293 (85.2)	1.32 (0.85, 2.05)	0.217		
Time on ART (years)						
<2	29 (26.4)	81 (73.6)	2.90 (1.70, 4.89)	<0.001	2.18 (1.08, 4.31)	0.027
2–5	18 (9.8)	168 (90.3)	0.87 (0.48, 1.52)	0.628	1.03 (0.54, 1.91)	0.934
>5 (ref)	44 (11.0)	356 (89.0)	1		1	
ART regimen type at enrollment						
NNRTI (ref)	38 (10.0)	344 (90.0)	1			
PI	51 (17.3)	243 (82.7)	1.90 (1.21, 3.00)	0.005	1.07 (0.67, 1.87)	0.809 ^2^
Integrase	4 (15.0)	23 (85.0)	1.57 (0.44, 4.36)	0.424	1.64 (0.41, 5.40)	0.442
Type NRTI ART at enrollment						
ABC (ref)	64 (13.0)	434 (87.0)	1			
AZT	19 (17.0)	94 (83.0)	1.37 (0.77, 2.36)	0.269		
TDF	10 (11.0)	83 (89.0)	0.82 (0.38, 1.59)	0.575		
ART regimen at 1st DRT						
NNRTI-based (ref)	38 (10.0)	344 (90.0)	1		1	
PI-based	51 (17.3)	244 (82.7)	1.89 (1.21, 2.99)	0.005	1.06 (0.61, 1.86)	0.831
Integrase-based	4 (14.8)	23 (85.2)	1.57 (0.44, 4.36)	0.424	1.65 (0.41, 5.42)	0.437
History of virologic failure						
No (ref)	27 (5.4)	477 (94.6)	1		1	
Yes	55 (38.2)	89 (61.8)	10.92 (6.60, 18.47)	<0.001	9.57 (5.61, 16.70)	<0.001
WHO stage						
1–2	67 (13.8)	420 (86.2)	1.27 (0.74, 2.30)	0.413		
3–4 (ref)	17 (11.2)	135 (88.8)	1			
**Clinic Volume**						
Heavy (ref)	55 (13.0)	366 (87.0)	1			
Medium	24 (15.0)	134 (85.0)	1.19 (0.70, 1.98)	0.507		
Light	14 (11.0)	111 (89.0)	0.84 (0.43, 1.53)	0.582		
**Clinic Location**						
Urban (ref)	55 (13.0)	366 (87.0)	1			
Peri-urban	24 (15.0)	134 (85.0)	1.19 (0.70, 1.98)	0.507		
Rural	14 (11.0)	111 (89.0)	0.84 (0.43, 1.53)	0.582		

^1^ Adjusted analysis of age group removing time on ART variable from the model; ^2^ adjusted analysis of ART regimen at enrollment removing regimen at first DRT.

**Table 4 viruses-15-02083-t004:** Clinical outcomes among children with virologic failure undergoing HIV drug resistance testing prior to 12-month study visit in intervention arm.

	Total Children with DRT Results *n* = 77 (%)	Children Recommended to Change ART after DRT Review *n* = 36 (%)	Children Recommended Not to Change ART after DRT Review ^1^ *n* = 31 (%)	Children Changing ART Due to Transition to DTG *n* = 10 (%)
12-month viral load result				
Suppressed	32 (42)	21 (58)	12 (39)	9 (90)
Not suppressed	21 (27)	10 (28)	10 (32)	1 (10)
Lost to follow up	9 (12)	3 (8)	6 (19)	0 (0)
Died	1 (1)	0 (0)	1 (3)	0 (0)
Missing	4 (5)	2 (6)	2 (6)	0 (0)

^1^ Two participants with an ART change recommendation prior to the final study visit did not change ART but are included. One was recommended to change from ABC/3TC/LPV to AZT/3TC/LPV but resuppressed without change. One was recommended to add raltegravir (RAL) to a PI-containing regimen due to triple-class resistance, but RAL was not available; patient resuppressed by 12-month visit.

## Data Availability

The de-identified data presented in this study are available on request from the corresponding author. The data are not publicly available due to the sensitive nature of the disease being studied.

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
