# Peer review of "HIV Drug Resistance Patterns and Characteristics Associated with Clinically Significant Drug Resistance among Children with Virologic Failure on Antiretroviral Treatment in Kenya: Findings from the Opt4Kids Randomized Controlled Trial"

_viruses, 2023, doi:10.3390/v15102083_

Round 1

Reviewer 1 Report

This study is of relevance since it reports on drug resistance in children living with HIV in a resource-limited setting by comparing standard of care with targeted drug resistance testing. They describe drug resistance patterns, characteristics associated with resistance, as well as the clinical outcomes after 12 months. 

The manuscript does, unfortunately, have many small errors that significantly detract from its quality. There are many instances where the denominator changes with no clear rationale provided, leading to the conclusion that it was an error. There are many small mathematical errors. Rounding is inconsistently done throughout the manuscript, with the authors sometimes giving no decimal, in other places 1 decimal and, in some tables, 2 decimals. The whole manuscript should be carefully edited before resubmission.

Abstract:

Abbreviation, VF, never introduced.

Introduction:

Abbreviation, DR, not introduced when used for the first time.

“The World Health Organization (WHO) and others have expressed concerns ….[4-6]” but references 4 to 6 are all from the WHO, so it is not clear who the “others” are.

“Among children with VF, small surveillance reports in east and southern Africa indicate between 48-71% may have DR mutations to non-nucleos(t)ide-reverse transcriptase inhibitor (NNRTI)-containing ART.[6-8]” Reference 6 is for the WHO report so not sure that this is the correct reference here? Should it perhaps rather be reference 9: Crowell CS, Maiga AI, Sylla M, Taiwo B, Kone N, Oron AP, et al. High Rates of Baseline Drug Resistance and Virologic Failure 454 Among ART Naive HIV-Infected Children in Mali. The Pediatric infectious disease journal. 2017. doi: 10.1097/INF.0000000000001575. 455 PubMed PMID: 28198788.

“Kenya is heavily burdened by HIV with an estimated 83,000 CHIV with around 60% 72 on ART in 2021.[14]” Reference 14 cannot be correct since it is from 2020 and the data cited are from 2021: “Joint United Nations Programme on HIV/AIDS. UNAIDS Kenya-HIV and AIDS Estimates 2020”.

Materials and methods:

Please add more context: “In brief, 704 CHIV ages 1-14 years were enrolled from five public facilities in Kisumu County, Kenya between March and December 2019 and followed for 12 months.” Please provide detail about whether these children were treatment naïve or experienced, as well as information about the CD4 and VL results at the start of the study. It is stated in the discussion: “We had very few children newly initiating ART and so are unable to estimate pre-treatment DR in this cohort.” The breakdown of the groups is important for interpreting the results.

The abbreviations for “point of care”, “viral load” and “antiretroviral” are not introduced when they are used for the first time.

How often did the Clinical Management Committee meet?

This reference style is out of keeping with the rest: “Genotypic Resistance Interpretation Algorithm (i.e., Stanford HIVdb) on any DRT.21”  

“the seven most common DR mutations (K65R, L74V/I, Y115F, M184V, 170 K103N, Y181C, G190A)” – how determined? A priori or in the dataset?

“We selected potential risk factors a priori, based on existing literature and content  knowledge, which included age, sex, duration on ART, prior antiretroviral exposure, prior history of VF”. What about other risk factors that have been associated with treatment failure, such as CD4 count and VL? “Given the exploratory nature of this analysis, we also included the following covariates: WHO stage, self-reported HIV status of primary caregiver at enrollment, clinic location urban, peri-urban, or rural and clinic volume defined as high, medium, and low based number of HIV patient visits per month at each facility.” It is not clear why “Household commodities ownership” is included 

Results:

Please correct the rounding so that the proportions add up to 100: “A total of 349 (49.5%) and 355 (50.4%)”

Please check the numbers: “including 24 (7.6%) control group participants”. 24/355 = 6.8%.

“Among intervention participants, 88 (25%) children experienced at least one episode of VF, and all had a least one DRT requested for a total of 166 DRT requests and 152 (92%) with results.” Please provide more information about how the results from participants with more than one test result were analysed, i.e. which results was entered in the tables and figures?

“Prior to the final study visit in the control group, 57 (16%) were identified with VF; but not included in our DRT analyses; 5 (9%) of these had a DRT requested through the standard of care procedures with 2 (40%) having results.” What is the reason for non-inclusion? The numbers are difficult to follow since, in figure 1, there were 66 participants with virological failure in the control group.  

“A total of 93 (88.0%) had major mutations and 13 (12.1%) had minor mutations only (Table 1).” The results shown in Table 1 are for “clinically significant drug resistance only”. It would be better to use the same terms throughout, especially since the definition of “clinically significant DR” is not only a major mutation: “if they had any mutation listed for NRTI and NNRTI drugs with a penalty score or if listed as “major” for PIs by Stanford’s Genotypic Resistance Interpretation Algorithm (i.e., Stanford HIVdb) on any DRT”.

Please be consistent with rounding: 93/106 = 87.7% and 13/106 = 12.3%.

It might be easier if the denominator is included every time, since this seems to be causing errors. For instance: “In children with major mutations, 87 (92.6%) had NNRTI resistance, 70 (74.5%) had NRTI, and 10 (10.9%) had PI resistance.” The denominator is 93, so all the percentages are incorrect. The same is true in the following section: “Additionally, more than half of children with major resistance had dual class resistance to NRTI and NNRTIs (n=51, 54.3%), two (2.1%) had dual class with either NRTI or NNRTI and PIs, and eight (8.5%) had triple class to NRTI, NNRTI, and PIs.”

“with most common being M184V (57.0%, 95% CI 48%,67%)” The percentage shown in Table 2 is 57.7%. Both instances are incorrect since  61/106 = 57.5%.

Please be specific with regards to the percentages: “with nearly half having DR to rilpivirine and fewer having DR to etravirine and doravirine.”

This paragraph is written twice – initially on page 7 and again on page 9. “Among the 88 CHIV in the intervention group of the study who had DRT requested, 47 (53%) had more than one DRT requested for repeat viremia during the study, whereas no CHIV in the control group had more than on DRT. Only three participants initially had minor resistance and later had major resistance on repeat DRT. Similarly, only six children had significant change in class of DR such as going from a single class with resistance to two or more drug classes. No children with change in resistance on follow up DRT had ART regimen changes between DRT.”

“Association between participant characteristics and clinically significant DR are  shown in Tables 1 and 3.” Only descriptions, and not associations, are shown in Table 1.

“Median age and sex were similar between the overall cohort and DR groups (Table 1).” No p-values are shown.

Please correct the percentages in these sentences: “In the intervention group, 38/88 (42.7%) were recommended to change ART after case review while 34 (42.5%) were advised to continue the current regimen. An additional 12 (13.5%)…”

“Nearly a third (N=11, 28.9%) with an ART change recommendation were able to preserve their current NRTI medications based on DRT results, which would have otherwise been switched per in-country guidelines.” This is a very important point: please be clear that the change recommended was for the non-NRTI component of the regimen.

Please add the numbers in the appropriate place: “Most children without an ART regimen change recommendation were on a PI-containing regimen 27/34 (79.4%) without…”

“In the intervention group, 38/88 (42.7%) were recommended to change ART after case review while 34 (42.5%) were advised to continue the current regimen. An additional 12 (13.5%) intervention participants underwent an ART change due to national transition to dolutegravir that was initiated by clinic providers.”

Why is the denominator 88 when the number of children with DR results is only 84 (as shown in Figure 1)?

It is also not clear why these numbers do not align with those presented in Table 4, since the table does include categories for missing VL results.

“For intervention participants with DRT prior to final study visit (N=77)…” As stated above, this should be 84 and, despite stating that 77 is used as the denominator, the number of cases add up to 84. All the percentages are therefore incorrect. The percentages manage to add up to 100% since 28/77 is incorrectly given as 27% and not 36%.

The abbreviation “VS” is never introduced.

Table 1:

Remove N(%) from the heading row since some of the values given in the table are medians and interquartile ranges.

Change “Integrase-based” to “INSTI-based”.

Double-check the rounding throughout the table e.g. 324/704 is not 46.1% but 46.0%.

Correct the alignment of the variables in the “Household commodities ownership” section.

Figure 1:

Remember to introduce the abbreviations used in the figure.

Please correct the numbers of the controls: 66 and 290 do not add up to 355.

I could not find footnote 2 in the figure.

Figure 2:

Section a does not have a number.

Some numbers in the titles are in brackets and others are not.

In section a, not all mutations have a percentage shown e.g. a) G190A and K103N.

It is very difficult to understand the percentages shown; e.g. in a) the total number is given as 106. Table 2 shows 61 M184V mutations, so it is not possible to understand how the percentage can be 21% in the figure. I suspect that the N is confusing since it denotes the number of children with drug resistance whereas the figure actually shows the proportion of  specific mutations among all the mutations detected.

Table 3:

Please correct the alignment of the p-values.

Remove the space: “0. 582”

Table 4:

N=77(%) – this is confusing since it seems to be a heading but an actual number is included.

The heading “Children without ART change recommendation after DRT review” suggests that these children had not received a recommendation whereas the recommendation was, in fact, not to change their regimen. It would therefore be better to align it with the other heading “Children recommended to change ART after DRT review” i.e. change to “Children recommended not to change ART after DRT review”.

Discussion:

“However, a similarly high proportion were found to have any DR (89%).” Correct the concord error. It should be clear to which study the percentage applies to. It should also be clear that this proportion is in children with virological treatment failure.

The abbreviation for LMIC is never introduced.

Remove the duplicated reference: “but as noted in our study and others,[6]those on PI-containing 381 regimens as second line were less likely to resuppress.[6]”

Please correct the spelling of both rilpivirine and cabotegravir: “rilpvirine” and “caobtegravir/rilpivrine”.

“Similar results in adults in South Africa prompted investigators to suggest that DRT may be required before long-acting injectable ART can be used in LMIC.[33, 34]”. Reference 34 is not from South Africa:

34. Cervo A, Russo A, Di Carlo D, De Vito A, Fabeni L, D'Anna S, et al. Long-acting combination of cabotegravir plus rilpivirine: A picture of potential eligible and ineligible HIV-positive individuals from the Italian ARCA cohort. J Glob Antimicrob Resist. 2023;34:141-4. Epub 2023/07/16. doi: 10.1016/j.jgar.2023.07.006. PubMed PMID: 37453495.

References:

Please correct the following references:

4. Organization WH. Global report on early warning indicators of HIV drug resistance. . Geneva, Switzerland: 2016.”

15. Health KMo. Report of the 2013 cross-sectional survey of acquired HIV drug resistance among adults and children on antiretroviral therapy at sentinel sites in Kenya. . In: Program NAaSC, editor. Nairobi, Kenya2016.” Please correct the author name.

20. Laboratories KNPH. National HIV Research Lab 2022 [cited 2023 August 8]. Available from: https://nphl.go.ke/national-hiv-480 reference-laboratory-nhrl.  

21. University S. HIV Drug Resistance Database 2021. Available from: https://hivdb.stanford.edu.

References 23 and 24 are the same:

23. Ministry of Health NASCPN. DTG Use of Dolutegravir based Regimen in Adolescent Girls & Women Living with HIV. Nairobi, Kenya: NASCOP, 2020 Print. 2020 Edition.

24. Ministry of Health NASCPN. Use of Dolutegravir based regimen in Adolescent Girls and Women Living with HIV Nairobi, Kenya: . Nairobi, Kenya.

Language editing would improve the quality of the manuscript. There are a number of grammatical errors. I list a few below:

“children had VF to first-line ART” – “to” should be “on”.

“protease inhibitors (PI)-containing ART” should be “protease inhibitor (PI)-containing ART”.

Please correct the grammar: “Children ages 1-14 years of age, enrolled at a study site, on or initiating ART, and with consenting caregiver were enrolled to the study.”

“HIV care and treatment was provided by government staff per national guidelines.”

Please correct the grammar: “within in two years prior to study enrolment”

“and low based number of HIV patient”

“We further estimate prevalence of”

Please correct the grammar: “Additional description of DR found by drug class are shown in Figure 2.”

Please add a full stop and correct the concord error: “(Figure 3) DR to newer generation NNRTIs were also high,..”

Please restructure this sentence for clarity: “However, younger children 1-4 years of age had over 3-fold odds of major DR compared to older children aged 5-10 years or 11-14 years demonstrated (odds ratio (OR) 3.2, 95% CI 1.6, 6.1).

Please correct the grammar: “Similarly, those on ART for < 2 years had and children…”

 The sentence does not have a subject: “In this cohort of over 700 children on ART in Kenya, detected high levels of DR in CHIV with VF.”

Reviewer 2 Report

In this manuscript, the authors analyzed the HIV drug resistance mutation data from a previous pediatric HIV treatment study. The authors performed detailed analysis of the HIV drug resistance mutations against the regimens the participants were using. They author’s main finding is that in patients with treatment failure, majority of them had major drug resistance mutations. Treatment history less than 2 years and previous treatment failure is associated with major drug resistance mutations. They author also demonstrated the data of re-suppression of HIV after adjustment of the treatment regimen.

Overall the paper is well-written and provides important data regarding how drug resistance mutations are developed in children with ART. I have a few comments and suggestions.

1. The authors need to include the ethical statements, including IRB, consent process, even though they were described in the previous publication.

2. Table 1, the rows for the regimens were mis-aligned.

3.  Any data showing the tolerance of LPV/r tolerance? The poor tolerance of LPV/r can contribute to

4. Line 199, why not included these patients. Can it introduce any bias?

5. Table 2. CI needs to be displayed in %, in consistent with the previous column.

6. A few comments on Figure 2. Figure 2a, how are dual class resistance show on the figure? Figure 2a, missing caption “a”, missing several %. Figure 2d, what are the meaning of the color coding

7. line 223, “on” should be “one”

8. 233 to 235, please summarize the how the mutation changes.

9. 253-299, duplicated from 230-237

10. Do the authors have any adherence data? The children with ART < 2 years had higher chance to develop resistance mutations. Do you think enhancing adherence for children who just initiate ART would prevent possible treatment failure and resistance?
